# HIERARCHICAL FORESIGHT: SELF-SUPERVISED LEARNING OF LONG-HORIZON TASKS VIA VISUAL SUBGOAL GENERATION

**Suraj Nair**[1,†]**, Chelsea Finn**[1,2]
[1]Stanford University, [2]Google Brain

## ABSTRACT

Video prediction models combined with planning algorithms have shown promise in enabling robots to learn to perform many vision-based tasks through only self-supervision, reaching novel goals in cluttered scenes with unseen objects. However, due to the compounding uncertainty in long horizon video prediction and poor scalability of sampling-based planning optimizers, one significant limitation of these approaches is the ability to plan over long horizons to reach distant goals. To that end, we propose a framework for subgoal generation and planning, hierarchical visual foresight (HVF), which generates subgoal images conditioned on a goal image, and uses them for planning. The subgoal images are directly optimized to decompose the task into easy to plan segments, and as a result, we observe that the method naturally identifies semantically meaningful states as subgoals. Across four simulated vision-based manipulation tasks, we find that our method achieves more than 20% absolute performance improvement over planning without subgoals and model-free RL approaches. Further, our experiments illustrate that our approach extends to real, cluttered visual scenes.

## 1 INTRODUCTION

Developing robotic systems that can complete long horizon visual control tasks, while generalizing to novel scenes and objectives, remains an unsolved and challenging problem. Generalization to unseen objects and scenes requires robots to be trained across diverse environments, meaning that detailed supervision during data collection in not practical to provide. Furthermore, reasoning over long-horizon tasks introduces two additional major challenges. First, the robot must handle large amounts of uncertainty as the horizon increases. And second, the robot must identify how to reach distant goals when only provided with the final goal state, a sparse indication of the task, as opposed to a shaped cost that implicitly encodes how to get there. In this work, we aim to develop a method that can start to address these challenges, leveraging self-supervised models learned using only unlabeled data, to solve novel temporally-extended tasks.

Model-based reinforcement learning has shown promise in generalizing to novel objects and tasks, as learned dynamics models have been shown to generalize to new objects (Finn & Levine, 2016; Ebert et al., 2018b), and can be used in conjunction with planning to reach goals unseen during training. However, planning to reach temporally distant goals is difficult. As the planning horizon increases model error compounds, and the cost function often provides only a noisy or sparse signal of the objective. Both of these challenges are exacerbated when planning in visual space.

In this work, the key insight that we leverage is that while model error and sparse cost signals can make long horizon planning difficult, we can mitigate these issues by *learning to break down long-horizon tasks into short horizon segments*. Consider, for example, the task of opening a drawer and putting a book in it, given supervision only in the form of the final image of the open drawer containing the book. The goal image provides nearly no useful cost signal until the last stage of the task, and model predictions are likely to become inaccurate beyond the first stage of the task. However, if we can generate a sequence of good subgoals, such as (1) the robot arm grasping the

---

[†]Work completed at Google Brain
Videos and code are available at: `https://sites.google.com/stanford.edu/hvf`

drawer handle, (2) the open drawer, and (3) the robot arm reaching for the book, planning from the initial state to (1), from (1) to (2), from (2) to (3), and from (3) to the final goal, the problem becomes substantially easier. The subgoals break the problem into short horizon subsegments each with some useful cost signal coming from the next subgoal image.

Our main contribution is a self-supervised hierarchical planning framework, hierarchical visual foresight (HVF), which combines generative models of images and model predictive control to decompose a long-horizon visual task into a sequence of subgoals. In particular, we propose optimizing over subgoals such that the resulting task subsegments have low expected planning cost. However, in the case of visual planning, optimizing over subgoals corresponds to *optimizing within the space of natural images*. To address this challenge, we train a generative latent variable model over images from the robot's environment and optimize over subgoals in the latent space of this model. This allows us to optimize over the manifold of images with only a small number of optimization variables. When combined with visual model predictive control, we observe that this subgoal optimization naturally identifies semantically meaningful states in a long horizon tasks as subgoals, and that when using these subgoals during planning, we achieve significantly higher success rates on long horizon, multi-stage visual tasks. Furthermore, since our method outputs subgoals conditioned on a goal image, we can use the same model and approach to plan to solve many different long-horizon tasks, even with previously unseen objects. We first demonstrate our approach in simulation on a continuous control navigation task with tight bottlenecks, and then evaluate on a set of four different multi-stage object manipulation tasks in a simulated desk environment, which require interacting with up to 3 different objects. In the challenging desk environment, we find that our method yields at least a 20% absolute performance improvement over prior approaches, including model-free reinforcement learning and a state of the art subgoal identification method. Finally, we show that our approach generates realistic subgoals on real robot manipulation data.

## 2 RELATED WORK

Developing robots that can execute complex behaviours from only pixel inputs has been a well studied problem, for example with visual servoing (Mohta et al., 2014; Espiau et al., 1992; Wilson et al., 1996; Yoshimi & Allen, 1994; Jagersand et al., 1997; Lampe & Riedmiller, 2013; Sadeghi et al., 2018; Sadeghi, 2019). Recently, reinforcement learning has shown promise in completing complex tasks from pixels (Ghadirzadeh et al., 2017; Levine et al., 2015; Kalashnikov et al., 2018; Lange et al., 2012; OpenAI et al., 2018; Schenck & Fox, 2016; Matas et al., 2018; James et al., 2017; Singh et al., 2019), including in goal-conditioned settings (Kaelbling, 1993; Schaul et al., 2015; Andrychowicz et al., 2017; Sadeghi et al., 2018; Sadeghi, 2019; Nair et al., 2018a). While model-free RL approaches have illustrated the ability to generalize to new objects (Kalashnikov et al., 2018) and learn tasks such as grasping and pushing through self-supervision (Pinto & Gupta, 2015; Zeng et al., 2018), pure model-free approaches generally lack the ability to explicitly reason over temporally-extended plans, making them ill-suited for the problem of learning long-horizon tasks with limited supervision.

Video prediction and planning have also shown promise in enabling robots to complete a diverse set of visuomotor tasks while generalizing to novel objects (Finn & Levine, 2016; Kalchbrenner et al., 2016; Boots et al., 2014; Byravan & Fox, 2016). Since then, a number of video prediction frameworks have been developed specifically for robotics (Babaeizadeh et al., 2017; Lee et al., 2018; Ebert et al., 2017), which combined with planning have been used to complete diverse behaviors (Nair et al., 2018b; Ebert et al., 2018b; Paxton et al., 2018; Xie et al., 2019). However, these approaches still struggle with long horizon tasks, which we specifically focus on.

One approach to handle long horizon tasks is to add compositional structure to policies, either from demonstrations (Krishnan et al., 2017; Fox et al., 2018), with manually-specified primitives (Xu et al., 2017; Huang et al., 2018), learned temporal abstractions (Neitz et al., 2018), or through model-free reinforcement learning (Sutton et al., 1999; Barto & Mahadevan, 2003; Bacon et al., 2016; Nachum et al., 2018; Levy et al., 2019). These works have studied such hierarchy in grid worlds (Bacon et al., 2016) and simulated control tasks (Nachum et al., 2018; Eysenbach et al., 2018; Levy et al., 2019) with known reward functions. In contrast, we study how to incorporate compositional structure in learned model-based planning with video prediction models. Our approach is *entirely self-supervised*, without motion primitives, demonstrations, or shaped rewards, and scales to vision-based manipulation tasks.

Classical planning methods have been successful in solving long-horizon tasks (LaValle, 2006; Choset et al., 2005), but make restrictive assumptions about the state space and reachability between states, limiting their applicability to complex visual manipulation tasks. Similarly, completing long horizon tasks has also been explored with symbolic models (Toussaint et al., 2019) and Task and Motion Planning (TAMP) (Kaelbling & Lozano-Perez, 2011; Srivastava et al., 2014). However, unlike these approaches our method requires no additional knowledge about the objects in the scene nor any predefined symbolic states. Recently, there have been several works that have explored planning in learned latent spaces (Kurutach et al., 2018; Ichter & Pavone, 2018; Watter et al., 2015; Srinivas et al., 2018). This has enabled planning in higher dimensional spaces, however these methods still struggle with long-horizon tasks. Furthermore, our hierarchical planning framework is agnostic to state space, and could directly operate in one of the above latent spaces.

A number of recent works have explored reaching novel goals using only self-supervision (Finn & Levine, 2016; Eysenbach et al., 2019; Kurutach et al., 2018; Wang et al., 2019; Jayaraman et al., 2019; Nair et al., 2018a). In particular, time-agnostic prediction (TAP) (Jayaraman et al., 2019) aims to identify bottlenecks in long-horizon visual tasks, while other prior works (Nair et al., 2018a; Finn & Levine, 2016) reach novel goals using model-free or model-based RL. We compare to all three of these methods in Section 5 and find that HVF significantly outperforms all of them.

## 3 PRELIMINARIES

We formalize our problem setting as a goal-conditioned Markov decision process (MDP) defined by the tuple $(\mathcal{S}, \mathcal{A}, p, \mathcal{G}, C, \lambda)$ where $s \in \mathcal{S}$ is the state space, which in our case corresponds to images, $a \in \mathcal{A}$ is the action space, $p(s_{t+1}|s_t, a_t)$ governs the environment dynamics, $\mathcal{G} \subset \mathcal{S}$ represents the set of goals which is a subset of possible states, $\mathcal{C}(s_t, s_g)$ represents the cost of being in state $s_t \in S$ given that the goal is $s_g \in \mathcal{G}$, and $\lambda$ is the discount factor. In practice, acquiring cost functions that accurately reflect the distance between two images is a challenging problem (Yu et al., 2019). We make no assumptions about having a shaped cost, assuming the simple yet sparse distance metric of $\ell_2$ distance in pixel space in all of our experiments. Approaches that aim to recover more shaped visual cost functions are complementary to the contributions of this work.

In visual foresight, or equivalently, visual MPC (Finn & Levine, 2016; Ebert et al., 2018b), the robot collects a data set of random interactions $[(s_1, a_1), (s_2, a_2), ..., (s_T, a_T)]$ from a pre-determined policy. This dataset is used to learn a model of dynamics $f_\theta(s_{t+1}, s_{t+2}, ..., s_{t+h}|s_t, a_t, a_{t+1}, ..., a_{t+h-1})$ through maximum likelihood supervised learning. Note the states are camera images, and thus $f_\theta$ is an action-conditioned video prediction model. Once the model is trained, the robot is given some objective and plans a sequence of actions that optimize the objective via the cross entropy method (CEM) (Rubinstein & Kroese, 2004). In this work, we will assume the objective is specified in the form of an image of the goal $s_g$, while CEM aims to find a sequence of actions that minimize the cost $\mathcal{C}$ between the predicted future frames from $f_\theta$ and the goal image. While standard visual foresight struggles with long-horizon tasks due to uncertainty in $f_\theta$ as the prediction horizon $h$ increases and sparse $\mathcal{C}$ for CEM to optimize, in the next section we describe how our proposed approach uses subgoal generation to mitigate these issues.

## 4 HIERARCHICAL VISUAL FORESIGHT

**Overview:** We propose hierarchical visual foresight (HVF), a planning framework built on top of visual foresight (Finn & Levine, 2016; Ebert et al., 2018b) for long horizon visuomotor tasks. We observe that when planning to reach a long horizon goal given only a goal image, standard planning frameworks struggle with (1) sparse or noisy cost signals and (2) compounding model error. Critically, we observe that if given the ability to sample possible states, there is potential to decompose a long horizon planning task into shorter horizon tasks. While this idea has been explored in classical planning (Betts, 2010), state generation is a significant challenge when dealing with high dimensional image observations.

One of our key insights is that we can train a deep generative model, trained exclusively on self-supervised data, as a means to sample possible states. Once we can sample states, we also need to evaluate how easy it is to get from one sampled state to another, to determine if a state makes for a good subgoal. We can do so through planning: running visual MPC to get from one state to another and measuring the predicted cost of the planned action sequence. Thus by leveraging the low-dimensional space of a generative model and the cost acquired by visual MPC, we can optimize over a sequence of subgoals that lead to the goal image. In particular, we can explicitly search in

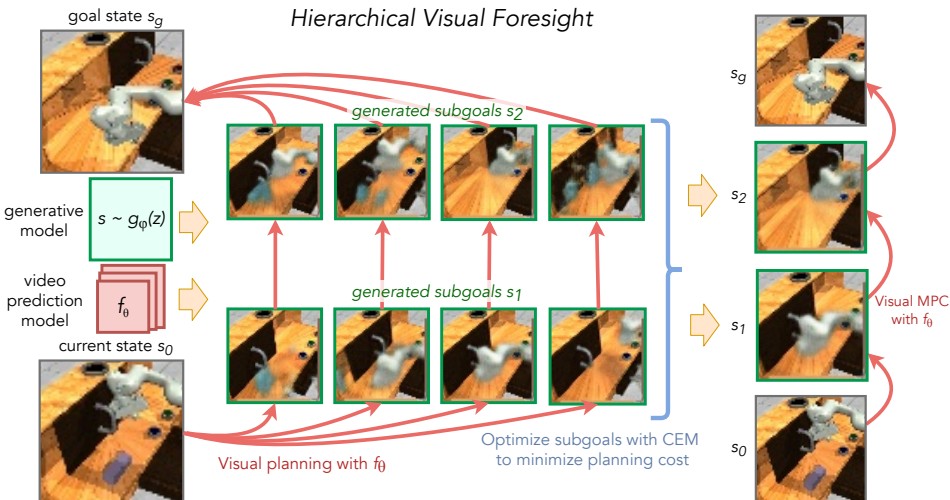

Figure 1: **Hierarchical visual foresight:** Our method takes as input the current image, goal image, an action conditioned video prediction model $f_\theta$, and a generative model $g_\phi(z)$. Then, it samples sets of possible states from $g_\phi(z)$ as sub-goals. It then plans between each sub-goal, and iteratively optimizes the sub-goals to minimize the worst case planning cost between any segment. The final set of sub-goals that minimize planning cost are selected, and finally the agent completes the task by performing visual model predictive control with the sub-goals in sequence. In this example the task is to push a block off the table, and close the door. Given only the final goal image, HVF produces sub-goals for (1) pushing the block and reaching to the handle and (2) closing the door.

latent image space for subgoals, such that no segment is too long-horizon, mitigating the issues around sparse costs and compounding model error.

In the following sections, we will describe more formally how HVF works, how we learn the generative model, how we learn the dynamics model, how goal-conditioned planning with visual MPC is executed, and lastly how subgoals are optimized.

**Hierarchical visual foresight:** Formally, we assume the goal conditioned MDP setting in Section 3 where the agent has a current state $s_0$, goal state $s_g$, cost function $\mathcal{C}$, and dataset of environment interaction $\{(s_1, a_1, s_2, a_2, ..., s_T, a_T)\}$. This data can come from any exploration policy; in practice, we find that interaction from a uniformly random policy in the continuous action space of the agent works well. From this data, we train both a dynamics model $f_\theta$ using maximum likelihood supervised learning, as well as a generative model $s \sim g_\phi$.

Now given $s_0$ and $s_g$, the objective is to find $K$ subgoals $s_1, s_2, ..., s_K$ that enable easier completion of the task. Our hope is that the subgoals will identify steps in the task such that, for each sub-segment, the planning problem is easier and the horizon is short. While one way to do this might be to find subgoals that minimize the total planning cost, we observe that this does not necessarily encourage splitting the task into shorter segments. Consider planning in a straight line: using any point on that line as a subgoal would equally minimize the total cost. Therefore, we instead optimize for subgoals that minimize the worst expected planning cost across any segment. This corresponds to the following objective:

$$\min_{s_1, ..., s_K} \max\{\mathcal{C}_{plan}(s_0, s_1), \mathcal{C}_{plan}(s_1, s_2), ..., \mathcal{C}_{plan}(s_K, s_g)\} \tag{1}$$

where $\mathcal{C}_{plan}(s_i, s_j)$ is the cost achieved by the planner when planning from $s_i$ to $s_j$, which we compute by planning a sequence of actions to reach $s_j$ from $s_i$ using $f_\theta$ and measuring the predicted cost achieved by that action sequence[1]. Once the subgoals are found, then the agent simply plans using visual MPC (Finn & Levine, 2016; Ebert et al., 2018b) from each $s_{k-1}$ to $s_k$ until a cost threshold is reached or for a fixed, maximum number of timesteps, then from $s_k^*$ to $s_{k+1}$, until planning to the goal state, where $s_k^*$ is the actual state reached when running MPC to get to $s_k$. For a full summary of the algorithm, see Algorithm 1. Next, we describe individual components in detail.

**Generative model:** To optimize subgoals, directly optimizing in image space is nearly impossible due to the high dimensionality of images and the thin manifold on which they lie. Thus, we learn a

---

[1]We compare max/mean cost in Section 5.4

---

**Algorithm 1** Hierarchical Visual Foresight HVF($f_\theta, g_\phi(z), s_t, s_g$)

---

1: Receive current state $s_t$ and goal state $s_g$
2: Initialize $q = \mathcal{N}(\mathbf{0}, \mathbf{I})$, $M = 200$, $M^* = 40$, number of subgoals $K$
3: **while** ($\sigma > 1e - 3$) or (iterations $< 5$) **do**
4:    **for** $m = 1, 2, ..., M$ **do**
5:       $s_0^m, s_{K+1}^m = s_t, s_g$
6:       $z^m \sim q$                                               # Sample latent subgoal lists
7:       **for** $k = 1, 2, ..., K$ **do**
8:          $s_k^m = g_\phi(z^m[k], s_0^m)$                             # Map latent to image subgoal
9:          $A_{plan,k}^m, C_{plan,k}^m = \text{MPC}(f_\theta, s_{k-1}^m, s_k^m)$       # Optimal actions and planning cost
10:       **end for**
11:       $A_{plan,k}^m, C_{plan,K+1}^m = \text{MPC}(f_\theta, s_K^m, s_{K+1}^m)$
12:       $Cost^m = \max_k \{C_{plan,0}^m, ..., C_{plan,K+1}^m\}$       # Max planning cost across segments
13:    **end for**
14:    $Z_{sort} = \text{Sort}(K : [Cost^1, .., Cost^M], V : [z^1, .., z^M])$       # Rank by $Cost^m$
15:    Refit $q$ to $\{Z_{sort}[1], ..., Z_{sort}[M^*]\}$ with low cost
16: **end while**
17: **for** $k = 1, 2, ..., K$ **do**
18:    Set final subgoal $s_k = g_\phi(Z_{sorted}[1]_k)$
19:    Execute $A_k, - = \text{MPC}(f_\theta, s_{k-1}, s_k)$
20: **end for**
21: Execute $A_K, - = \text{MPC}(f_\theta, s_K, s_g)$

---

generative model $s \sim g_\phi(z)$ such that samples represent possible futures in the scene, and optimize in the low-dimensional latent space $z$. In settings where aspects of the scene or objects change across episodes, we only want to sample plausible future states, rather than sampling states corresponding to all scenes. To handle this setting, we condition the generative model on the current image, which contains information about the current scene and the objects within view. Hence, in our experiments, we use either a variational autoencoder (VAE) or a conditional variational autoencoder (CVAE), with a diagonal Gaussian prior over $z$, i.e. $z \sim \mathcal{N}(\mathbf{0}, \mathbf{I})$. In the case of the CVAE, the decoder also takes as input an encoding of the conditioned image, i.e. $s \sim g_\phi(z, s_0)$. In practice, we use the inital state $s_0$ as the conditioned image. We train the generative model on randomly collected interaction data. The architecture and training details can be found in Appendix A.3

**Dynamics model:** The forward dynamics model $f_\theta$ can be any forward dynamics model that estimates $p(s_{t+1}, s_{t+2}, ..., s_{t+h}|s_t, a_t, a_{t+1}, ..., a_{t+h-1})$. In our case states are images, so we use an action conditioned video prediction model, stochastic variational video prediction (SV2P) (Babaeizadeh et al., 2017) for $f_\theta$. We train the dynamics model on randomly collected interaction data. Architecture and training parameters are identical to those used in (Babaeizadeh et al., 2017); details can be found in Appendix A.4.

**MPC & planning with subgoals:** When optimizing subgoals, we need some way to measure the ease of planning to and from the subgoals. We explicitly compute this expected planning cost between two states $s_k, s_{k+1}$ by running model predictive control $A, C = \text{MPC}(f_\theta, s_k, s_{k+1})$[2] as done in previous work (Finn & Levine, 2016; Nair et al., 2018b; Ebert et al., 2018b). This uses the model $f_\theta$ to compute the optimal trajectory between $s_k$ and $s_{k+1}$, and returns the optimal action $A$ and associated cost $C$. Note this does not step any actions in the real environment, but simply produces an estimated planning cost. Details on this procedure can be found in Appendices A.1 and A.2. Once the subgoals have been computed, the same procedure is run $\text{MPC}(f_\theta, s_{k-1}, s_k)$ until $s_k$ is reached (measured by a cost threshold) or for a fixed, maximum number of timesteps, then from $\text{MPC}(f_\theta, s_k^*, s_{k+1}), ..., \text{MPC}(f_\theta, s_K^*, s_g)$ (Alg. 1 Lines 17:21), where $s_k^*$ represents the state actually reached after trying to plan to $s_k$. In this case, the best action at each step is actually applied in the environment until the task is completed, or the horizon runs out. Note we only compute subgoals once in the beginning calling $\text{HVF}(f_\theta, g_\phi(z), s_0, s_g)$, the plan with those subgoals. In principle HVF can be called at every step, but for computational efficiency we only call HVF once.

**Subgoal optimization:** Since we need to search in the space of subgoals to optimize Equation 1, we perform subgoal optimization using the cross entropy method (CEM) (Rubinstein & Kroese, 2004) in the latent space of our generative model $g_\phi(z)$. Note this subgoal optimization CEM is distinct

---

[2]Pseudocode for MPC can be found in Appendix A.1

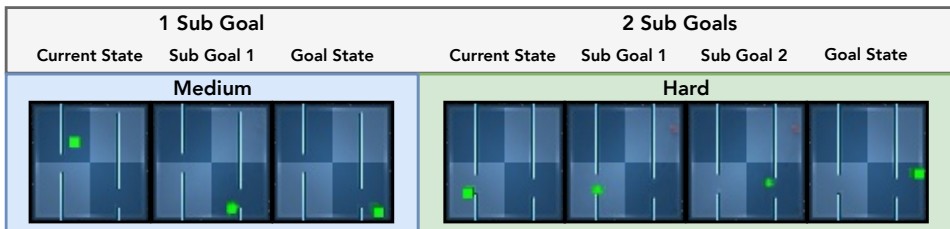

Figure 3: **Qualitative Results for Maze Navigation.** Here we show some of the generated subgoals from HVF for the Maze Navigation task. On the left we observe samples for $K = 1$ and on the right we observe samples for $K = 2$. We see in the "Medium" difficulty and $K = 1$, the generated subgoal is very close to the intuitive bottleneck in the task. Similarly, in the "Hard" case with $K = 2$ we observe that the discovered subgoals are close to the gaps in the walls, which represent the bottlenecks in the task.

from the CEM used for computing the planning cost, which we use as a subroutine within this outer level of CEM. At a high-level, the subgoal optimization samples a list of subgoals, evaluates their effectiveness towards reaching the goal, and then iteratively refines the subgoals through resampling and reevaluation. We begin by drawing $M$ samples from a diagonal Gaussian distribution $q = \mathcal{N}(\mathbf{0}, \mathbf{I})$ where the dimensionality of $q$ is $K * L$, where $K$ is the number of subgoals and $L$ is the size of the latent dimension $z$ of the generative model $g_\phi(z)$ (Alg. 1 Line 6). Thus, one sample $z$ from $q$ gives us $K$ latents each of size $L$, each of which is decoded into a subgoal image (Alg. 1 Line 8). Then, as described in Equation 1, the cost for one sample (one set of $K$ subgoals) is computed as the maximum planning cost across the subsegments (Alg. 1 Lines 9:12). The $M$ samples are ranked by this cost, and then $q$ is refit to the latents $z$ of the best $M^*$ samples (Alg. 1 Lines 14:15). In all of our experiments we use $M = 200$, $M^* = 40$ and $L = 8$.

## 5 EXPERIMENTS

In our experiments, we aim to evaluate **(1)** if, by using HVF, robots can perform challenging goal-conditioned long-horizon tasks from raw pixel observations more effectively than prior self-supervised approaches, **(2)** if HVF is capable of generating realistic and semantically significant subgoal images, and **(3)** if HVF can scale to real images of cluttered scenes. To do so, we test on three domains: simulated visual maze navigation, simulated desk manipulation, and real robot manipulation of diverse objects. The simulation environments use the MuJoCo physics engine (Todorov et al., 2012). We compare against three prior methods: (a) *visual foresight* (Finn & Levine, 2016; Ebert et al., 2018b), which uses no subgoals, (b) *RIG* (Nair et al., 2018a) which trains a model-free policy to reach generated goals using latent space distance as the cost, and (c) visual foresight with subgoals generated by *time-agnostic prediction (TAP)* (Jayaraman et al., 2019), a state-of-the-art method for self-supervised generation of visual subgoals, which generates subgoals by predicting the most likely frame between the current and goal state.

### 5.1 MAZE NAVIGATION

First, we study HVF in a 2D maze with clear bottleneck states, as it allows us to study how HVF compares to oracle subgoals. In the the maze navigation environment, the objective is for the agent (the green block) to move to a goal position, which is always in the right-most section. To do so, the agent must navigate through narrow gaps in walls, where the position of the gaps, goal, and initial state of the agent are randomized within each episode. The agent's action space is 2D Cartesian movement, and the state space is top down images. We consider three levels of difficulty, "Easy" where the agent spawns in the rightmost section, "Medium" where the agent spawns in the middle, and "Hard" where the agent spawns in the leftmost section. Details in Appendix B.1.

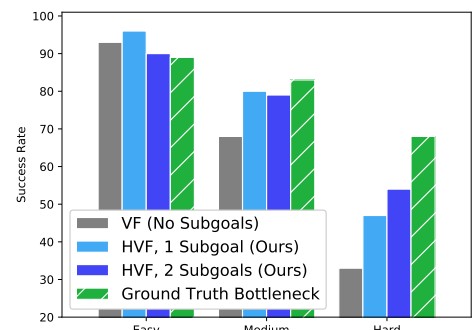

Figure 2: **Quantitative Results for Maze Navigation.** Success rate for navigation tasks using no subgoals, HVF subgoals, and ground truth hand specified subgoals. We observe that HVF significantly improves performance on the "Medium" and "Hard" difficulty compared to not using subgoals. Computed over 100 randomized trials.

Videos/code are available at: `https://sites.google.com/stanford.edu/hvf`

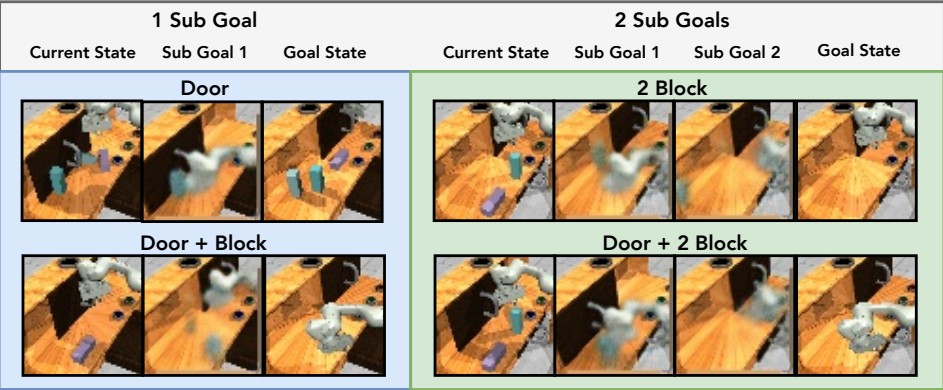

Figure 5: **Qualitative Results for Desk Manipulation.** Example generated subgoals from HVF for the desk manipulation tasks with one or two subgoals. We observe interesting behavior: in the Door Closing + Block Pushing task with one subgoal, the subgoal is to slide the door then push the block, while in the Door Closing + 2 Block Pushing the first subgoal is to push the blocks, then grasp the door handle, then slide the door.

**Results:** In Figure 2, we observe that using HVF subgoals consistently improves success rate over visual foresight (Finn & Levine, 2016) without subgoals, indicating that it is able to find subgoals that make the long-horizon task more manageable. Additionally, we compare to the "Ground Truth Bottleneck" that uses manually-designed subgoals, where the subgoal is exactly at the gaps in the walls (or the midpoint between states in the "easy" case). We find that while using the oracle subgoals tends to yield the highest performance, the oracle subgoals do not perform perfectly, suggesting that a non-trivial amount of performance gains are to be had from improving the consistency of the video prediction model and cost function for short-horizon problems, as opposed to the subgoals. We also show that HVF outperforms time agnostic prediction (TAP) (Jayaraman et al., 2019) in Appendix C.3.

We next qualitatively examine the subgoals discovered by HVF in Figure 3, and find empirically that they seem to correspond to semantically meaningful states. In this task there are clear bottlenecks - specifically reaching the gaps in the walls and find that HVF outputs close to these states as subgoals. For example, when the agent starts in the leftmost section, and has to pass through two narrow gaps in walls, we observe the first subgoal goal corresponds to the agent around the bottleneck for the first gap and the second subgoal is near the second gap.

## 5.2 SIMULATED DESK MANIPULATION

We now study the performance improvement and subgoal quality of HVF in a challenging simulated robotic manipulation domain. Specifically, a simulated Franka Emika Panda robot arm is mounted in front of a desk (as used in (Lynch et al., 2019)). The desk consists of 3 blocks, a sliding door, three buttons, and a drawer. We explore four tasks in this space: door closing, 2 block pushing, door closing + block pushing, and door closing + 2 block pushing. Example start and goal images for each task are visualized in Figure 5, and task details are in Appendix B.2. The arm is controlled with 3D Cartesian velocity control of the end-effector position. Across the 4 different tasks in this environment, we use a single dynamics model $f_\theta$ and generative model $g_\phi(z)$. Experimental details are in the Appendix B.2.

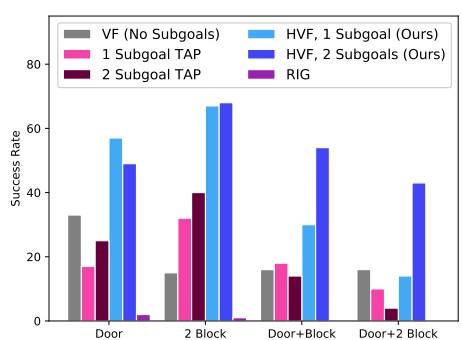

Figure 4: **Quantitative Results for Desk Manipulation:** Using HVF drastically improves performance. Across all 4 tasks, HVF with two subgoals leads to a more than 20% absolute performance improvement over visual foresight (Finn & Levine, 2016), TAP (Jayaraman et al., 2019), and RIG (Nair et al., 2018a). Computed over 100 trials with random initial scenes.

**Results:** As seen in Figure 4, we find that using HVF subgoals dramatically improves performance, providing at least a 20% absolute improvement in success rate across the board. In the task with the longest horizon, closing the door and sliding two blocks off the table, we find that using no subgoals or 1 subgoal has approximately 15% performance, but 2 subgoals leads to over 42% success rate. We compare to subgoals generated by time agnostic prediction (TAP) (Jayaraman et al., 2019) and find that while it does generate plausible subgoals, they are very close to the start or goal, leading to no

benefit in planning. We also compare against RIG (Nair et al., 2018a), where we train a model free policy in the latent space of the VAE to reach "imagined" goals, then try and reach the unseen goals.

However, due to the complexity of the environment, we find that RIG struggles to reach even the sampled goals during training, and thus fails on the unseen goals. Qualitatively, in Figure 5, we observe that HVF outputs meaningful subgoals on the desk manipulation tasks, often produces subgoals corresponding to grasping the handle, sliding the door, or reaching to a block.

## 5.3 REAL ROBOT MANIPULATION

Lastly, we aim to study whether HVF can extend to real images and cluttered scenes. To do so, we explore the qualitative performance of our method on the BAIR robot pushing dataset (Ebert et al., 2018a). We train $f_\theta$ and $g_\phi(z, s_0)$ on the training set, and sample current and goal states from the beginning and end of test trajectories. We then qualitatively evaluate the subgoals outputted by HVF. Further implementation details are in the Appendix B.3. **Results:** Our results are illustrated in Fig. 6. We observe that even in cluttered scenes, HVF produces meaningful subgoals, such grasping objects which need to be moved. For more examples, see Figure C.2. We observe that in both the BAIR dataset and the desk manipulation ex-

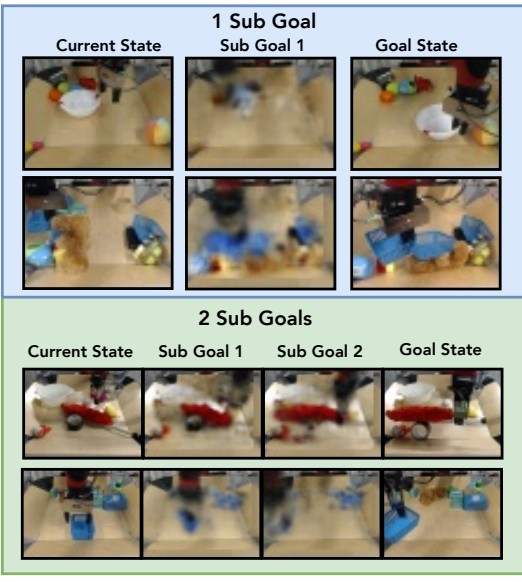

Figure 6: **BAIR Dataset Qualitative Results.** The subgoals generated by HVF on the BAIR dataset, which we find correspond to meaningful states between the start and goal. For example, when moving objects we see subgoals corresponding to reaching/grasping.

periments, the most common failure cases corresponded to the subgoal prediction and model ignoring the objects and focusing exclusively on the arm. Cost functions which can more effectively capture objects and their poses would be a step towards addresing this.

## 5.4 ABLATIONS

In our ablations, we explore three primary questions: **(1)** What is the optimal number of subgoals?, **(2)** Is there a difference between HVF using the max and mean cost across subsegments?, and **(3)** Is HVF as valuable when the samples used for visual MPC is significantly increased? We evaluate these questions in the maze navigation task on the "Hard" difficulty. All results report success rates over 100 trials using random initialization/goals. Additional ablations on planning horizion and latent space cost can be found in Appendix C.1.

| # subgoals | 0 | 1 | 2 | 3 | 5 | 10 |
|---|---|---|---|---|---|---|
| success | 33% | 47% | **54%** | 39% | 2% | 0% |

Table 1: **Number of Subgoals.** With a fixed sampling budget, as the number of subgoals increases beyond 2, performance drops as the subgoal search is challenging.

**Number of Subgoals:** We explore how HVF performs as we increase the number of subgoals in Table 1. Interestingly, we observe that as we scale the number of subgoals beyond 2, the performance starts to drop, and with 5 or more subgoals the method fails. We conclude that this is due to the increasing complexity of the subgoal search problem: the sampling budget allocated for subgoal optimization is likely insufficient to find a large sequence of subgoals.

| # subgoals | 1 | 2 |
|---|---|---|
| mean cost | 0.45 | 0.53 |
| max cost | **0.47** | **0.54** |

Table 2: **Max vs Mean.** Success rates for minimizing mean vs. max cost across subsegments. Max is marginally better.

**Max vs Mean:** In our HVF formulation, we define the subgoal optimization objective as finding the subgoals that minimize the max cost across subsegments. Table 2 compares to using the mean cost. We find that using the max cost is marginally better.

| # subgoals | 0 | 1 | 2 |
|---|---|---|---|
| 200 samples | 0.33 | 0.47 | 0.54 |
| 1000 samples | **0.35** | **0.54** | **0.55** |

Table 3: **Sample Quantity.** Success using visual MPC with 200 vs. 1000 samples at each CEM iteration. Using more samples is better, but HVF still outperforms standard visual foresight by a wide margin.

**Sample Quantity:** In Table 3, we examine how the number of action samples affects the relative improvement of HVF. Using more samples should also mitigate the challenges of sparse costs, so one might suspect that HVF would be less valuable in these settings. On

the contrary, we find that HVF still significantly outperforms no subgoals, and the improvement between using 0 and 1 subgoals is even more significant.

## 6 Conclusion and Limitations

We presented an self-supervised approach for hierarchical planning with vision-based tasks, hierarchical visual foresight (HVF), which decomposes a visual goal into a sequence of subgoals. By explicitly optimizing for subgoals that minimize planning cost, HVF is able to discover semantically meaningful goals in visual space, and when using these goal for planning, perform a variety of challenging, long-horizon vision-based tasks. Further, HVF learns these tasks in an entirely self-supervised manner without rewards or demonstrations.

While HVF significantly extends the capabilities of prior methods, a number of limitations remain. First, HVF assumes access to an exploration policy which can cover enough of the state space to learn a good model. While the random policy used works for our environments, more complex environments may require better exploration techniques, such as intrinsic motivation methods.

Another limitation of the current framework is its computational requirements, as the nested optimization procedure requires many iterations of MPC with expensive video prediction models. Specifically, assume that the horizon of the task is $T$, one iteration of visual MPC has computational cost $C$, and HVF is using $K$ subgoals, and searching in a space of $N$ subgoal sequences. Then normal visual foresight or TAP would have cost $T \times C$, while HVF would have cost $(T \times C) + (K \times N \times C)$. However note - the computation of $(K \times N \times C)$ can be heavily parallelized because it is done completely offline (without environment interaction). We also expect that this can be mitigated by reformulating the subgoal prediction problem as policy inference, and training a subgoal generation policy in the loop of training, an interesting direction for future work.

Further, while the development of accurate visual cost functions and predictive models were not the main aim of this work, the performance with oracle subgoals suggests this to be a primary bottleneck in furthering the performance of HVF. Specifically, learning dynamics models and cost functions which can more effectively capture the state of all the objects in the scene would significantly improve the performance of HVF.

Lastly, development of better generative models that can capture the scene and generate possible futures remains an open and challenging problem. Work in generative models that enable better generation of cluttered scenes with novel objects could improve the applicability of HVF in real-world settings.

### Acknowledgments

We would like to thank Sudeep Dasari, Vikash Kumar, Sherry Moore, Michael Ahn, and Tuna Tokosz for help with infrastructure, as well as Sergey Levine, Abhishek Gupta, Dumitru Erhan, Brian Ichter, and Vincent Vanhouke for valuable discussions. This work was supported in part by an NSF GRFP award. Chelsea Finn is a CIFAR Fellow in the Learning in Machines & Brains program.

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

## A  Method Details

### A.1  Visual MPC

In HVF, evaluating the cost of potential subgoals, as well as actually taking actions given computed subgoals uses visual MPC. One step of visual MPC is described in Algorithm 1. Given the current state and goal state, MPC samples action trajectories of length $H$, then feeds them through the model $f_\theta$. Then the cost of the output images are computed relative to the goal image, which is then used to sort the actions and refit the action distribution. After 5 iterations or convergence the best action is returned.

---

**Algorithm 2** MPC$(f_\theta, s_t, s_g)$

---

    Receive current state $s_t$ and goal state $s_g$ from environment
    Initialize $N(\mu, \sigma^2) = N(0, 1)$, cost function $C(s_i, s_j)$
    **while** $(\sigma^2 > 1e - 3)$ or (iterations $\leq 5$) **do**
        $a_1, ..., a_D \sim N(\mu, \sigma^2)$
        $s_{t+H,1}, ..., s_{t+H,D} = f_\theta(s_t, a_1, ..., a_D)$
        $l_1, ..., l_D = [C(s_{t+H,1}, s_g), ..., C(s_{t+H,D}, s_g)]$
        $a_{sorted} = Sort([a_1, .., a_D])$
        Refit $N(\mu, \sigma^2)$ to $a_{sorted}[1 - D^*]$
    **end while**
    Return $l_{sorted}[1], a_{sorted}[1]$
where $D = 200, D^* = 40$

---

### A.2  Planning with Subgoals

In the main text we describe how given, $s_0$ and $s_g$, HVF produces subgoal images $s_1, s_2, ..., s_K$. Given these subgoals, planning with them is executed as follows. For an episode starting at $s_0$ of maximum length $T$, the agent plans using visual MPC from $s_0$ to $s_1$ until some cost threshold $C(s_t, s_1) < x$ or for some maximum number of steps $T^*$. Once this criteria is reached, the agent plans from its current state to the next subgoal $s_2$, again until the cost threshold $C(s_t, s_2) < x$ or some maximum number of steps $T^*$, and so until the agent is planning form $s_t$ to the true goal $s_g$, at which point it simply plans until the environment returns that the task has been a success or the total horizon $T$ runs out.

### A.3  Generative Model

The generative model we use is either a Variational Autoencoder $s \sim g_\phi(z)$ or a Conditional Variational Autoencoder $s \sim g_\phi(z, s_0)$.

In the VAE, training is done by sampling images from the dataset, and each image is encoded using an image encoder into the mean and standard deviation of a normal distribution of dimension 8. Then a sample from the distribution are decoded back to the original input image. This is trained with a maximum likelihood loss as well as a KL penalty on the normal distribution restricting it to be a unit gaussian $\mathcal{N}(0, 1)$.

In the conditional VAE, training is done by sampling pairs of images from the dataset, specifically images from random episodes $s_t$ as well as the corresponding first image of that episode $s_0$. Then both $s_0$ and $s_t$ are encoded using the same image encoder, which are then flattened and concatenated. This is then encoded into the mean and standard deviation of a normal distribution of dimension 8. A sample from the resulting distribution is then fed through fully connected layers and reshaped into the output shape of the encoder, concatenated with the spatial embedding of the conditioned image from the encoder and then decoded. This is done to enable easier conditioning on the scene in the decoding. The resulting image is trained with a maximum likelihood loss as well as a KL penalty on the normal distribution restricting it to be a unit gaussian $\mathcal{N}(0, 1)$.

In the Simulated Maze Navigation experiments, we use a Conditional VAE which encodes the 64x64 RGB image with 4 convolutional layers ([16, 32, 64, 128] filters, kernel size 3x3, stride 2), as well as 3 fully connect layers of size 256 for the generated image and 3 fully connected layers of size [512, 512, 256] for the conditioned image, and a decoder with the same architecture as the encoder inverted. In the Simulated Desk Manipulation experiments, we use a VAE which encodes the 64x64

RGB image with 7 convolutional layers ([8, 16, 32, 32, 32, 64, 128] filters, kernel size 3, stride alternating between 1 and 2), as well as a single fully connected layer of size 512 before mapping the latent distribution, which is decoded with the inverted encoder architecture. In the Real Robot Manipulation experiments we use a Conditional VAE which encodes the 48x64 RGB image with 9 convolutional layers ([8, 16, 32, 64, 64, 128, 128, 256, 512] filters, with kernel size 3, and stride sizes [1,1,1,1,1,2,2,2]), as well as 3 fully connect layers of size 256 for the generated image and 3 fully connected layers of size [512, 512, 256] for the conditioned image, and a decoder which is the inverted architecture of the encoder. The three experiments use Adam optimizer with learning rate 1e-4, 1e-3, 1e-4 respectively.

## A.4 Dynamics Model

The dynamics model is an action conditioned video prediction model, stochastic variational video prediction (SV2P) (Babaeizadeh et al., 2017). See architecture below:

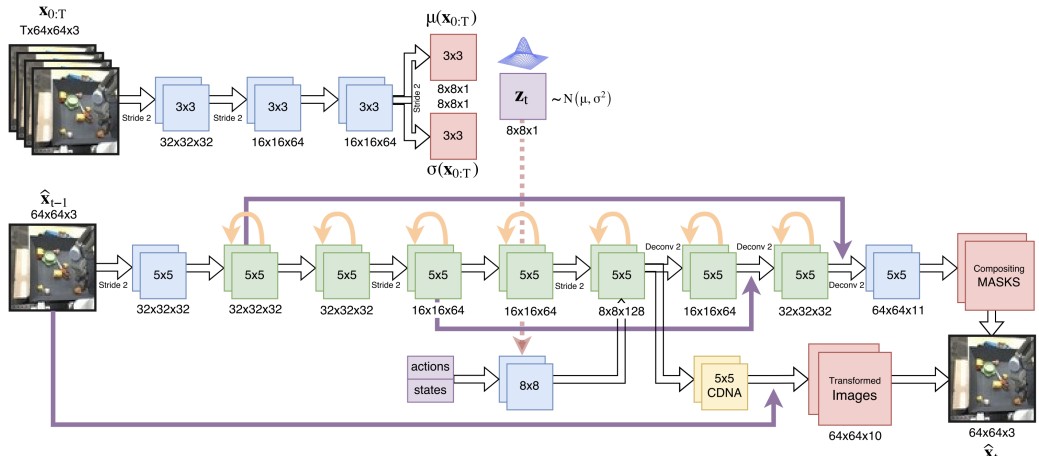

Figure 7: **Architecture of stochastic variational video prediction** model (SV2P) (Babaeizadeh et al., 2017) used for dynamics model $f_\theta$ (taken from (Babaeizadeh et al., 2017) with permission). The architecture has two main sub-networks, one convolutional network which approximates the distribution of latent values give all the frames, and a recurrent convolutional network which predicts the pixels of the next frame, given the previous frame, sampled latent, and action (if available). The code is open sourced in Tensor2Tensor (Vaswani et al., 2018) library.

For all experimental settings the dynamics models are trained for approximately 300K iterations.

## B Experiment Details

### B.1 Maze Navigation

**Data Collection:** For the maze navigation environment data is collected through random policy interaction. That is, for each action we sample uniformly in the delta $x, y$ action space of the green block. We collect 10000 episodes, each with random initialization of the walls, block, and goal. Each episode contains 100 transitions. The images are of size 64x64x3.

**Cost Function:** In this experiment, the cost function $C(s_i, s_j)$ is simply the squared $\ell_2$ pixel distance between the images, that is $||s_i - s_j||_2^2$

**Planning Parameters:** When planning in the maze navigation environment, MPC samples action trajectories of length $H = 5$. Additionally, the cost of a sequence of 5 actions is the cost of the last of the subsequent frames, where the cost of each frame is the temporal cost $C$ described above. Lastly, we use $T = 50$ and $T^* = 10$ in these experiments.

### B.2 Simulated Desk Manipulation

**Task Details:** *Door:* The first task is reaching to and closing the sliding door. From the initialization position of the arm, it needs to reach around the door handle into the correct position, then slide the

door shut. The position of the door and distractor blocks on the table are randomized each episode. The agent is given 50 timesteps to complete the task. *2 Block:* The second task requires the agent to push two different blocks off the table, one located on the left end of the table and one located in the middle. It requires pushing first one block, then re positioning to push the other block off the table. The agent is given 50 timesteps to complete the task. *Door + Block:* The agent must both slide the door closed from a random position, as well as push a block off of the table. This requires both grasping and sliding the door from the previous task, as well as positioning the end effector behind the block to slide it off the table. The agent is given 100 timesteps to complete the task. *Door + 2 Block:* A combination of the 2 Block and Door + Block task. The agent must close the door and slide both blocks off the table within 100 timesteps.

**Data Collection:** For the simulated desk manipulation environment data is again collected through random policy interaction. That is, for each action we sample uniformly in the delta $x, y, z$ action space of the robot end effector. We collect 10000 episodes, each with random initialization of the desk door/drawer and blocks. Each episode contains 100 transitions. The images are of size 64x64x3.

**Cost Function:** In this experiment, the cost function $C(s_i, s_j)$ is simply the squared $\ell_2$ pixel distance between the images, that is $||s_i - s_j||_2^2$

**Planning Parameters:** When planning in the desk manipulation environment MPC samples action trajectories of length $H = 15$, which actually consists of 5 actions, each repeated 3 times. Additionally, the cost of a sequence of 15 actions is the $\ell_2$ pixel cost $C$ of the **last frame only**. Note this is distinct from the maze environment. Lastly, we use $T = 50$ or $100$ and $T^* = 20$ in these experiments, depending on the task.

### B.3 Real Robot Manipulation

**Data:** We use the BAIR robot manipulation dataset from (Ebert et al., 2018a), which consists of roughly 15K trajectories split into a train/test split. We train $f_\theta$ and $g_\phi$ on the train set and display qualitative results on the test set.

**Cost Function:** Like the desk manipulation set up, the cost function $C(s_i, s_j)$ is the squared $\ell_2$ pixel distance between the images, that is $||s_i - s_j||_2^2$

**Planning Parameters:** When planning in the desk manipulation environment MPC samples action trajectories of length $H = 15$, which actually consists of 5 actions, each repeated 3 times. Additionally, the cost of a sequence of 15 actions is the $\ell_2$ pixel cost $C$ of the **last frame only**.

## C Additional Results

### C.1 Additional Ablations

#### C.1.1 Planning Horizon

In the maze environment, on the hard difficulty, we study how increasing the planning horizon of visual MPC impacts the benefit of using HVF (See Table 4). Interestingly, we find that for longer planning horizons, performance does not necessarily improve (as a longer planning horizon constitutes a harder search problem). This supports the idea that simply using longer planning horizon is not necessarily the solution to doing long-horizon tasks. Further, we find that even with longer planning horizons, HVF outperforms standard visual foresight, but that performance with 2 subgoals is worse than with 1 subgoal.

#### C.1.2 VAE Latent Space Cost

To address the sparsity of pixel cost, we explore using distance in the latent space of the VAE as

| # SG | 0 | 1 | 2 |
|---|---|---|---|
| **5 Step Planning** | 0.33 | 0.47 | **0.54** |
| **10 Step Planning** | 0.46 | **0.55** | 0.37 |
| **15 Step Planning** | 0.31 | **0.39** | 0.24 |

Table 4: **Planning Horizon.** Compares success rates using visual MPC with planning horizon of 5,10,and 15. Using subgoals always performs the best, but for larger planning horizons 2 subgoals can hurt performance.

| | Easy | Medium | Hard |
|---|---|---|---|
| **VAE Latent $\ell_2$ Cost** | 0.62 | 0.49 | 0.23 |
| **Pixel $\ell_2$ Cost** | **0.93** | **0.68** | **0.33** |

Table 5: **Latent vs Pixel $\ell_2$ cost.** Compares success rates using visual MPC with either the squared $\ell_2$ pixel distance or squared $\ell_2$ distance in the latent space of the VAE. The latent space cost is notably worse due to it providing a weak signal at close distances.

a cost signal. However, we find that this cost actually performs worse across all difficulties. In Table 5 we show the success rate of standard visual foresight for either pixel $\ell_2$ cost or VAE latent space $\ell_2$ cost. We find that the reason for the poor performance of the latent space cost is that it provides close to zero cost anytime the green blocks are reasonably close together, leading to CEM often getting stuck close to the goal but not actually at the goal.

## C.2 Real Robot Manipulation Examples

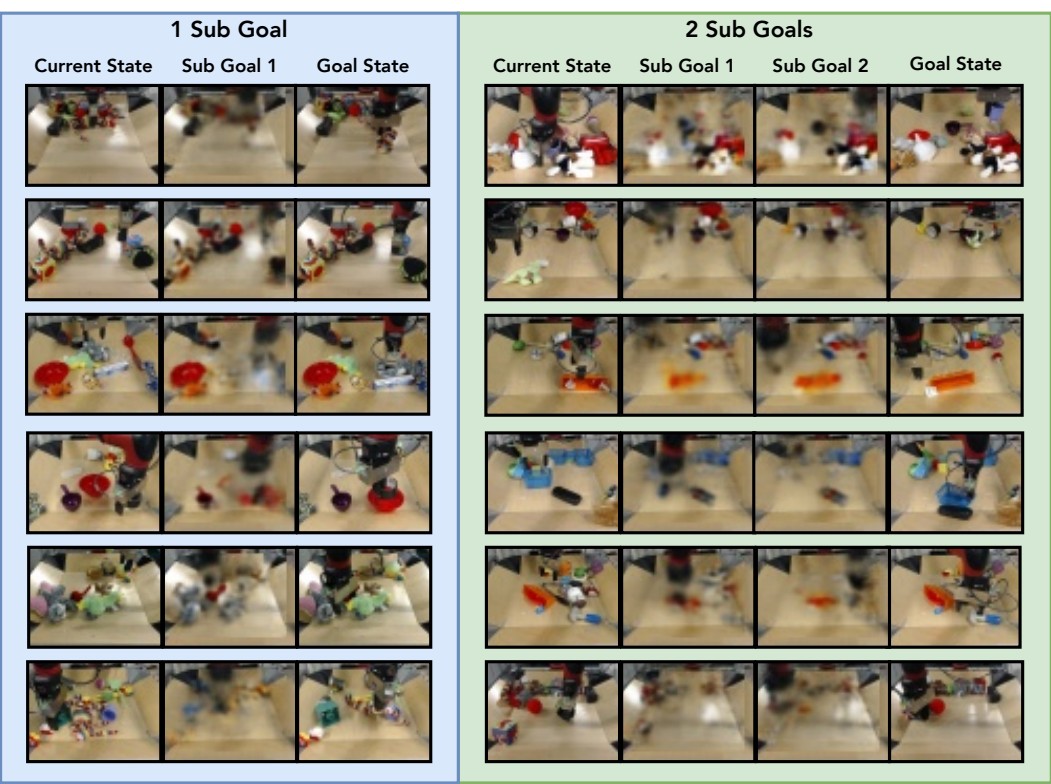

Figure 8: **BAIR Dataset Additional Results.**

## C.3 Time Agnostic Prediction(Jayaraman et al., 2019) in the Maze Navigation Task

We observe that while TAP gets similar performance to HVF on the "Easy" and "Medium" cases, it has significantly lower performance in the longest horizon "Hard" setting.

Additionally we see that Recursive TAP with 2 subgoals has lower performance across all difficulties as the subgoals it outputs are very close to current/goal state (See Figure 9).

| Difficulty | Easy (1 SG) | Medium | Hard |
|---|---|---|---|
| **TAP (1 SG)** | 0.89 | **0.82** | 0.35 |
| **TAP (2 SG)** | 0.91 | 0.68 | 0.28 |
| **HVF (1 SG)** | **0.96** | 0.80 | 0.47 |
| **HVF (2 SG)** | 0.90 | 0.79 | **0.54** |

Table 6: **TAP on Maze Environment.** Compares success rates of TAP vs HVF on the maze environment. Computed over 100 randomized trials.

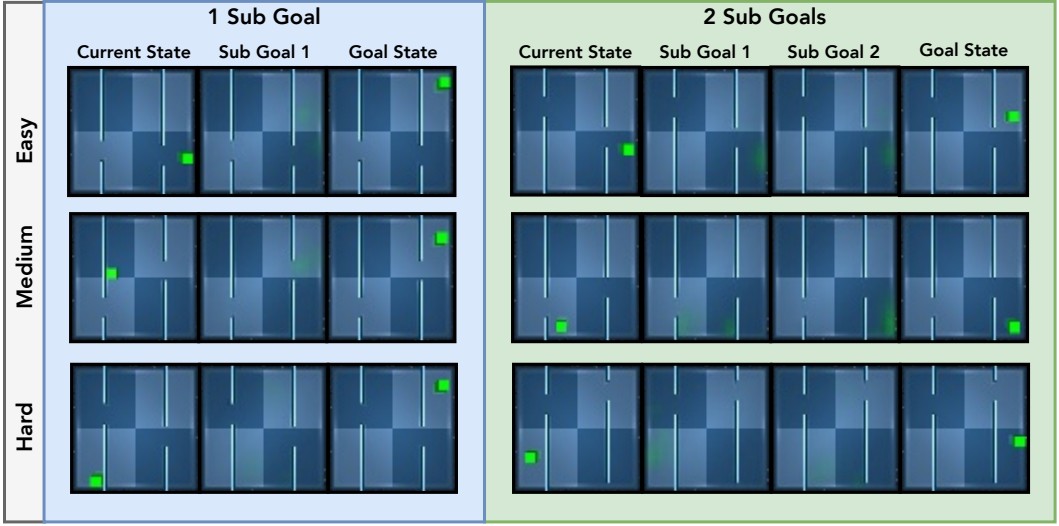

Figure 9: **Qualitative examples of TAP.** Notice it is prone to predicting very close to the current or goal state for subgoals.

