# OpenReview forum: "Hierarchical Foresight: Self-Supervised Learning of Long-Horizon Tasks via Visual Subgoal Generation"
_ICLR.cc/2020/Conference — Accept (Poster)_

### Official Review · AnonReviewer1 · 2019-10-14
**Official Blind Review #1**

**Rating:** 6

**Review:**

This paper introduces a hierarchical extension to existing work in vision-based model predictive control. Here, a hierarchical model is optimised to find sub-goals that minimise the planning cost (bottleneck states), so as to allow for improved planning to goal states expressed in higher dimensional state spaces. As expected, results show that this hierarchy improves tasks execution success rates.

The paper is clearly written, although many sub-components of the architecture are described in other articles. While I appreciate that the work is incremental, and that it relies on a number of previously modules, this brevity and the architectural complexity means that the work will be challenging to reproduce or adapt to different tasks.

Nevertheless, I believe the idea is interesting, and the paper does achieve its stated goals of allowing for entirely self-supervised learning without motion primitives, demonstrations and rewards, so makes a useful contribution here.  Compositional planning through sub-goals is a good idea, and this paper adds a useful mechanism to identify suitable sub-goals in a latent space and plan through these.

However, I don't believe that this approach is in any way practically feasible at present, and the paper makes a number of hyperbolic claims that should be toned down to give a more balanced perspective (I appreciate that the authors attempted to do so in their limitations section, but this needs to be done throughout the paper).

Specifically:

1.) Please avoid the use of relative improvement percentages (200% performance improvements in the abstract - the 20% absolute performance is strong enough to stand alone).

2.) The paper oversimplifies the importance of exploration and data collection in the proposed approach. The paper states that a uniform random policy in the continuous action space of the agent works well, and offsets the data collection to "any exploration policy", but this is a key requirement if the proposed approach is to be useful. At present the method seems to be relying on typewriting monkeys writing Shakespeare as a precursor to control. An exploration policy using identified sub-goal of interest may be a particularly valuable extension in future work, but for now the paper glosses over this aspect.

3.) The use of the term "long-horizon tasks" and "manipulation". In line with point 2, my personal feeling is that any task or state that can conceivably be explored and accessed by uniformly sampling from a continuous action space does not qualify as a long-horizon task. From the results and videos, it seems that all of the tasks could be solved by following a single trajectory in 3D space, ie. swing left to knock objects off the table, then right to close the door. Terming this "long-horizon" is a bit of a stretch, as is the notion that flopping about a table bashing into objects counts as "manipulation". Motion planning and trajectory following are not long-term horizon manipulation tasks, solving towers of Hanoi is. Moreover, I appreciate that experiments were conducted on real robot datasets, but this seems to be more of an exercise in latent space anthropomorphism than practical evidence of a feasible control policy.

4.) Limitations regarding latent space expressiveness.  As mentioned in the limitations section of the paper, the proposed approach is heavily reliant on a latent space that fully captures the scene and can roll-out future states sensibly. This is an extremely challenging problem, and one which appears to affect the proposed approach, for example, in the accompanying video (3:.40 - 3:59), the desired goal state contains two objects standing on the table and a closed door, but the policy only closes the door, while objects are knocked over. This seems to indicate that the latent space and planning is unable to learn good object embeddings and spatial representations, and that the type of task that can be solved is along the lines of move forward and to the right when the black blob is on the left of the image, or move down to the left, when there are some coloured blobs on the table.

5.) Limitations around goal state representations . Following on from 4, the identification of important aspects in a given image or latent space represents a major challenge to the proposed approach. Given an image of a desired state, how can the proposed approach identify which elements in the scene are required for success, and which are simply distractors? At present, I see no mechanism by which this could ever be learned in a self-play setting or through an image-based goal state. Eg. what if the image indicated that I want the two objects to be in a specific position, and I never cared about the state of the door?

Despite these limitations, the granularity of the solution is fine for a proof of concept work like this, and is itself a commendable achievement. Unfortunately, the paper's choice of language, relying on terms like "long term horizon" and "manipulation" for simple reaching and pushing tasks exaggerates the state of robot learning, and is potentially misleading to those less familiar with the field.

Despite my gripes, the paper definitely meets the ICLR threshold of "accept if you'd share it with a colleague", and I believe it is is a useful approach to sub-goal identification and a nice piece of work, so I recommend acceptance.



**Experience Assessment:**

I have published one or two papers in this area.

**Review Assessment: Checking Correctness Of Derivations And Theory:**

I assessed the sensibility of the derivations and theory.

**Review Assessment: Checking Correctness Of Experiments:**

I assessed the sensibility of the experiments.

**Review Assessment: Thoroughness In Paper Reading:**

I read the paper at least twice and used my best judgement in assessing the paper.

---

> ### Author Response · Authors · 2019-11-09
> **Response to Review #1**
>
> We thank reviewer 1 for their thoughtful comments. We address each comment below:
>
> (1): Thank you for pointing this out - we have switched to only using absolute percentages in the new revision.
>
> (2): We agree that the ability to explore the environment well is a critical assumption of this line of work, and this may not hold for more complex environments. The depth of literature in exploration [1,2,3] can likely be combined with HVF in a straightforward matter. We added a discussion of this to the limitations section in the new revision.
>
> (3): We modified the introduction of the paper to use these terms less frequently. Regardless of the semantics of the terminology “long-horizon”, we empirically find that existing methods, including both model-free and model-based RL methods, struggle to solve the tasks due to their time horizon and sparse reward signal, while HVF enables better performance by reducing the horizon. Regarding “manipulation”, we agree that we are not doing dexterous/fine grained manipulation, but more coarse object pushing/door sliding. We are open to rephrasing to “robotic control” if the reviewer finds that term preferable.
>
> (4): Learned dynamics models and cost functions from images do have inherent limitations. We have added further discussion of this in the limitations section of the paper.
>
> (5): This is a great point - and indicates a core challenge in using goal images to specify task. In reality there are only some parts of the image that matter, and the others can vary significantly. Prior work has studied how to combine visual foresight with classifiers that ignore irrelevant factors [4]. An approach such as [4] could be combined with HVF in a straight-forward manner, to measure a more semantic distance to an image of the goal.
>
> [1] Deepak Pathak, Dhiraj Gandhi, and Abhinav Gupta. Self-supervised exploration via disagreement. In Proceedings of the 36th International Conference on Machine Learning, 2019.
> [2] Yuri Burda, Harrison Edwards, Amos Storkey, and Oleg Klimov. Exploration by random network distillation. arXiv preprint arXiv:1810.12894, 2018.
> [3] A. Gupta, R. Mendonca, Y. Liu, P. Abbeel, and S. Levine. Meta-reinforcement learning of structured exploration strategies. arXiv preprint arXiv:1802.07245, 2018.
> [4] A. Xie, A. Singh, S. Levine, and C. Finn, “Few-shot goal inference for visuomotor learning and planning,” Conference on Robot Learning (CoRL), 2018.

---

### Official Review · AnonReviewer2 · 2019-10-24
**Official Blind Review #2**

**Rating:** 6

**Review:**

This paper proposes a method, hierarchical visual foresight (HVF) that learns to break down the long horizon tasks into short horizon segments. It first generates the subgoals conditioned on the main goal. These subgoals are optimized to have meaningful states and used for planning. The experiments on Maze navigation, simulated desk manipulation, and real robot manipulation show significant performance gain over the planning method without subgoals and model-free RL.

The paper tackles a novel and challenging problem for the long-horizon tasks. Also, the paper is well written and easy to understand.

Questions/Concerns:

- How important is the output quality of the video prediction model?

- Finding subgoals that split into the optimal subtasks seems not easy. Are the learned subgoals actually reasonable for the final task? I can see that Fig. 6 has some examples of subgoals but is hard to recognize what are the subtasks.
- Also, is there a chance that subgoals are randomly selected? Could you verify that subgoals are consistent across different initialization/runs?
- Could you provide us any failure cases if there's any and explain us why this happens?

- The number of subgoals needs to be fixed. I assume this number depends on how complex of the task is. It is surprising that only 2 subgoals were enough for the BAIR Robot push dataset (Tab. 1). The authors commented that "the sampling budget allocated for subgoal optimization is likely insufficient to find a large sequence of subgoals.". Will it actually be solved by increasing the sampling? Or is it possible that the dataset is too simple or simply finding optimal subgoals is hard.  Could authors comment more about this issue?
- Also, how can you find the right number of subgoals?

- The computational requirement is one of the weaknesses. Please provide the comparison of how much is the computational cost of the proposed method compared to other ones such as VF (without subgaols), TAP, and RIG.


**Experience Assessment:**

I have published one or two papers in this area.

**Review Assessment: Checking Correctness Of Derivations And Theory:**

I did not assess the derivations or theory.

**Review Assessment: Checking Correctness Of Experiments:**

I assessed the sensibility of the experiments.

**Review Assessment: Thoroughness In Paper Reading:**

I read the paper thoroughly.

---

> ### Author Response · Authors · 2019-11-09
> **Response to Review #2**
>
> We thank reviewer 2 for their comments. We address each comment below:
>
> “Could you provide us any failure cases and explain why this happens?”: We added a discussion of this to Section 5.3 of the revision. The most common failure case is that the visual MPC cost focused too heavily on the arm, and as a result produced plans (and subgoals) which ignore the object changes which are a critical part of the task. One solution to this would be better cost functions which provide a denser, more object relevant cost.
>
> Comparison of Computation Cost: We added a detailed discussion of computational cost to the last section of the paper, as follows: Assume that the horizon of the task is T, one iteration of visual MPC has cost C, and HVF is using K subgoals, and searching in a space of N subgoal sequences. Then normal VF/TAP would have cost T*C, while HVF would have cost (T*C) + (K*N*C). However note - the computation of (K*N*C) can be completely parallelized because it is done completely offline (without environment interaction). RIG is significantly more costly in terms of environment interaction, since it learns a model free policy, but evaluating the policy is cheap — a single feedforward model.
>
> “How important is the output quality of the video prediction model?”: The quality of the short-term predictions is important for HVF to achieve good performance. For visual MPC *without* HVF to work well, the video predictions must be high quality much further into the future. The gap between oracle subgoals (<70%) and perfect performance (100%) is an indication of how much prediction quality is limiting the performance of HVF.
>
> “Are the learned subgoals reasonable for the final task?”: We qualitatively find that in our tasks the predicted subgoals do seem reasonable for the given tasks. For example in Figure 3, we see that the predicted subgoals map to close to the gaps in the walls, which are bottlenecks in the task. Similarly, in the bottom left of Figure 5, we see that when the task involves sliding the door shut and pushing a block off the table, the generated subgoal corresponds to first sliding the door shut. Additional examples on the BAIR dataset are also in Figure 8 in the appendix.
>
> “Could you verify that subgoals are consistent across different initialization/runs?”: Because there are multiple possible solutions of good subgoals we observe that the subgoals differ a small amount depending on the initialization, as in any stochastic optimization. However the general semantic subgoal is consistent (e.g. grasp the door handle), even if there is slight variation in the images. Regarding better than random performance, our results indicate stronger performance than Time-Agnostic Prediction, a state of the art subgoal prediction method, over 100 random initial states.
>
> “It is surprising that only 2 goals were enough for the BAIR robot pushing dataset”: The optimal number of subgoals does depend on the complexity of the task, particularly the horizon (as opposed to visual complexity). So despite the visual complexity of the BAIR dataset, 2 subgoals still should effectively break at task into 3 sub segments as observed.
>
> “How can you find the right number of subgoals?”: Extending the sampling-based optimization over subgoals to also optimize over the number of subgoals is a relatively straight-forward extension of HVF, which we leave for future work.

---

### Author Response · Authors · 2019-09-30
**Anonymized Code Link**

We realized that the original code link was not fully-anonymized. We have disabled sharing on that link. Here is a fully-anonymous link: https://drive.google.com/file/d/1cEQtdVmsYMRr3R0QFrClVNT-9BcgYBWF/view?usp=sharing

---

### Decision · Program_Chairs · 2019-12-19

**Decision:**

Accept (Poster)

**Comment:**

This paper proposes a method that uses subgoals for planning when using video prediction. The reviewers thought that the paper was clearly written and interesting. The reviewer questions and concerns were mostly addressed during the discussion phase, and the reviewers are in agreement that the paper should be accepted.